# Understanding the challenges of healthcare transition in the context of HIV-related stigma for young adults with perinatal HIV in Thailand

Linda Aurpibul[1]*, Chutima Saisaengjan[2], Wipaporn Natalie Songtaweesin[2,3,4‡], Supunnee Masurin[1], Tulathip Suwanlerk[5], Thanyawee Puthanakit[2,3‡]

1 Research Institute for Health Sciences, Chiang Mai University, Chiang Mai, Thailand, 2 Center of Excellence in Pediatric Infectious Diseases and Vaccines, Chulalongkorn University, Bangkok, Thailand, 3 Department of Pediatrics, Faculty of Medicine, Chulalongkorn University, Bangkok, Thailand, 4 School of Global Health, Faculty of Medicine, Chulalongkorn University, Bangkok, Thailand, 5 TREAT Asia/amfAR – The Foundation for AIDS Research, Bangkok, Thailand

☉ These authors contributed equally to this work.
‡ WNS and TP also contributed equally to this work.
* linda.a@cmu.ac.th

## Abstract

The healthcare transition during adolescence and young adulthood has consistently been reported as a critical period for attrition and adverse health outcomes. The study assessed HIV-related stigma and transition experiences among young people living with perinatal HIV (YPHIV) in Thailand. We conducted a mixed-methods cross-sectional study at two research sites in Chiang Mai and Bangkok, Thailand from December 2023 to February 2024. We recruited YPHIV aged between 18–30 years who remained under care in pediatric HIV clinics (group A), those who had transitioned to adult care from those clinics (group B), and caregivers of group B participants (group C). We assessed HIV-related stigma using the validated 8-item Thai Internalized HIV-related Stigma Scale brief (Thai-IHSS brief) and transition-related experiences through in-depth interviews and focus group discussions. Thirty YPHIV (median age 23 years [IQR 22–26]) and ten caregivers were enrolled. The Thai IHSS brief score revealed a low level of internalized stigma in the study participants (median score 14; IQR 11–17). Anticipated negative thoughts and negative self-thoughts were common. HIV-related stigma experiences of YPHIV and caregivers were grouped into 3 themes: internalized, anticipated, and enacted stigma/discrimination. Transition experiences of YPHIV in both groups included hesitation to navigate care in adult clinics and feeling unprepared, perceived loss of support, and demotivation from being in care. Anticipated stigma and social problems were expressed by YPHIV and caregivers. In the focus groups, YPHIV indicated their need to learn about the transition beforehand, to be guided to the new clinic while staying connected to their original clinics, and to extend time in the pediatric clinic until they were more confident with transitioning care. In conclusion, we found many stigma

**Data availability statement:** Public availability would compromise patient privacy. We uploaded a dataset containing anonymous quantitative data as S1 File. We also shared some quotes in S2 File. However, the whole qualitative data set contains potentially identifying and sensitive patient information in most of the texts as transcribed from the in-depth interview session. As we have only few participants, sharing with the public may risk the identification of study participants who are among the vulnerable populations. External request for data access can be sent to either the corresponding author or the research administrative department of the Research Institute for Health Science, Chiang Mai University at the E-mail: research-rihes@cmu.ac.th.

**Funding:** This study was supported by grants to amfAR from ViiV Healthcare, and the U.S. National Institutes of Health's National Institute of Allergy and Infectious Diseases, the Eunice Kennedy Shriver National Institute of Child Health and Human Development, the National Cancer Institute, the National Institute of Mental Health, the National Institute on Drug Abuse, the National Heart, Lung, and Blood Institute, the National Institute on Alcohol Abuse and Alcoholism, the National Institute of Diabetes and Digestive and Kidney Diseases, and the Fogarty International Center, as part of the International Epidemiology Databases to Evaluate AIDS (IeDEA; U01AI069907). LA and WNS are supported in part through a grant from amfAR, The Foundation for AIDS Research, with support from the US National Institutes of Health's Fogarty International Center and the National Institute of Mental Health (CHIMERA; D43TW011302). This work is solely the responsibility of the authors and does not necessarily represent the official views of any of the institutions mentioned above. This publication is the result of funding in whole or in part by the NIH. It is subject to the NIH Public Access Policy. Through acceptance of this federal funding, NIH has been given the right to make this manuscript publicly available in PubMed Central upon the Official Date of Publication, as defined by NIH.

**Competing interests:** The authors have declared that no competing interests exist.

issues started since childhood, plus collective experience while growing up. The internalized HIV-related stigma influenced the healthcare transition journey of YPHIV. Healthcare providers need additional guidance on how to manage transition in YPHIV, including individualized transition plans for those at increased risk of adverse outcomes, interventions to manage internalized stigma, and follow-up strategies after transition.

## Introduction

With wide access to effective antiretroviral therapy (ART), HIV infection is now considered a chronic health condition, people with HIV can live extended lives and children born with perinatal HIV can grow to adulthood [1,2]. High rates of loss-to-follow-up, treatment failure, and mortality observed in young people with perinatal HIV after transitioning out of pediatric HIV clinics [3–5] highlight significant transition-related challenges. As of 2023, an estimated 1800 children living with HIV were under 15 years of age in Thailand, most of them having grown up and incorporated into the 570,000 people living with HIV (PLWH) between 15–49 years of age. The healthcare transition period during adolescence and young adulthood has consistently been reported as a critical period [6]. Not all are able to self-manage and navigate adult clinics smoothly after transition, and current services may not offer what is needed to engage them in care. A study in Bangkok reported decreased retention rates from 92% at 1 year to 87% at 2 years post-transition [7]. Mental health problems, lack of perceived social support, as well as HIV-related stigma were identified as barriers to engagement in care, treatment adherence, and transition to adult care [8,9].

Studies from other regions reveal many transition-related challenges including difficulty navigating adult HIV clinics, long waiting times, loss of informational support, lack of privacy, changing providers, and fear of stigma [10,11]. Retention rates of only 50–75% were observed after 1–4 years of transition, and worsened retention was observed between older adolescence and young adulthood [11,12]. HIV-related stigma was reported as a factor attributable to ART non-adherence, poor engagement and retention in care, and unsuccessful transition [13]. With limited qualitative data from the Asia-Pacific region, it is challenging to understand the perspective of service users in this context. This study aimed to assess HIV-related stigma and transition experiences of young people with perinatal HIV (YPHIV) in Thailand to inform the design of interventions to facilitate effective transition.

## Methods

### Study settings and population

A multiple methods cross-sectional study was conducted at two research sites in Chiang Mai and Bangkok, Thailand, from December 2023 to February 2024. We have similar cohorts of YPHIV who were initiated on HIV treatment and followed during 2005–2020 as a part of the TREAT Asia Pediatric HIV Observational Database [14]. In 2021, there were 124 and 145 YPHIV aged 18 years or older actively followed at



the sites in Chiang Mai and Bangkok, respectively; their treatment outcomes during adolescent years were included in the previous publication [15]. This study was composed of two parts. The first was a questionnaire and an in-depth interview (IDI) to explore individual experiences around stigma and transition. Potential participants were approached by study staff in the clinics during their regular visit for HIV care or reached out by phone for those who had transitioned out. Before enrolling in the study, those who expressed interest underwent an informed consent process. We purposely recruited YPHIV aged between 18 and 30 years with a known history of vertical transmission who remained under care in pediatric HIV clinics (group A), YPHIV who had transitioned to adult care from those clinics (group B), and caregivers of participants who remained in the pediatric clinics (group C). The second part was a focus group discussion (FGD), with one group at each site, to create a space for YPHIV to share their transition experiences and brainstorm on their transition needs. We invited participants who underwent IDIs in either Group A or B who were interested in participating on another separate day by appointment.

## Data collection

The demographic and socioeconomic status of participants were collected at study entry. The variables collected were sex, age, duration on ART, and the highest level of education. The most recent CD4 and viral load, self-reported adherence, and the Patient Health Questionnaire (PHQ-9) scores for depression, which were assessed on every clinic visit, were obtained from medical records. The 8-item Thai Internalized HIV-related Stigma Scale brief (Thai-IHSS brief) was used to assess internalized stigma. It is a tool developed from survey data among Thai PLWH and had a congruence validity and discriminant power in measuring internalized stigma [16]. The 4-point rating scale (strongly disagree, disagree, agree, strongly agree) was used to assess negative thoughts and their effects towards self, family, and access to healthcare services. It rated stigma levels as low (8–15), medium (16–23), and high (24–32).

An interview guide was developed after a literature review and discussion between investigators with working experiences with YPHIV around transition. Topics covered at in-depth interviews were: transition experiences (physical, mental, and psychosocial effects); social support/family involvement/other influencers; needs for transition (preparation, training, support, monitoring, trial period, buddy system, long-term engagement with groups/clinics, etc.); disclosure issues; and HIV-related stigma issues (perceived, anticipated, and experienced). The interviews took place in an undisturbed room in each clinic. All interviewers were healthcare providers trained in qualitative interviews with clinical experience working with YPHIV. All interviews were audio-recorded with participants' permission. Each interviewer wrote notes after the interview session ended. During the biweekly study meeting, interviewers from both sites presented and discussed their findings. Interviews were conducted until data saturation.

FGD participants were recruited from IDI participants. Interested participants provided their consent. Each group was composed of 7 YPHIV. The participants and group facilitators (LA, CS), with one note taker and one study coordinator at each site, were present during the discussion in a clinic meeting room. Discussions lasted 90–135 minutes, including group activities at the beginning, which helped to elicit their thoughts and ideas on the topics while listening and sharing with others. We followed the step-by-step guide for FGD [17]. The FGD guide was prepared before the study started and revised after IDI analysis with the aim of further exploring YPHIV's experiences and needs. Topics included how to better prepare for healthcare transition and organize youth-friendly services in the existing healthcare system to minimize attrition of YPHIV after the transition. Participants were reimbursed $30 and $25 for IDI and FGD, respectively.

## Data analysis

Qualitative data from IDI and FGD audio files were transcribed verbatim. Transcripts were read line-by-line by two research team members, then shared and discussed between the interviewer and another team member. Deductive-focused analysis was applied before a codebook was developed and used to code all transcripts. A theme matrix was constructed based on coded data. Excerpts were divided into HIV-related stigma, healthcare transition experiences, and

transitional needs. Excerpts were organized under major themes and subthemes. Dedoose Software (Version 9.2.6) was used for data analysis. All interview transcript analyses were in Thai. Selected excerpts were translated into English during manuscript preparation to support results in each theme.

### Ethics considerations

This study was approved by the Human Experimentation Committee, Office of Research Ethics of the Research Institute for Health Sciences, Chiang Mai University (certificate of approval no. 70/2023), and the Institutional Review Board, Faculty of Medicine, Chulalongkorn University (certificate of approval no. 1479/2023). Written informed consent was obtained from each participant before enrollment.

## Results

We enrolled 30 YPHIV: 17 (57%) were female; the median age was 23 years (IQR 22–26). The median duration of ART was 18 years (IQR 15–21). The latest CD4 count was 583 cells/mm³ (IQR 331–757). Self-reported adherence was 95% (IQR 80–100%), and 27 (90%) were virologically suppressed at the most recent assessment. Ten caregivers were enrolled (n = 5/site). Their characteristics are shown in Table 1.

### HIV-related stigma

**Internalized stigma.** The median Thai IHSS brief score was 14 (IQR 11–17), representing a low level of internalized stigma in the overall study participants. The components with the highest scores indicating their most concern were anticipated negative thoughts ("Others may end their relationship with me if they learn that I am living with HIV" and "Others will think it serves me right if they know I am living with HIV") and negative thoughts toward self ("I am ashamed that I am living with HIV" and "I think that I have HIV because of my bad karma").

**Table 1. Characteristics of young people with perinatal HIV (YPHIV) and caregivers who participated in the study.**

| Characteristics | YPHIV (n = 30) | Caregivers* (n = 10) |
|---|---|---|
| Female sex | 17 (57%) | 9 (90%) |
| Age (years) | 23 (22-26) | 53 (47-56) |
| Highest level of education | | |
| Primary school | 2 (6.7%) | 1 (10%) |
| Secondary school | 7 (23%) | 3 (30%) |
| High school | 4 (13%) | 2 (20%) |
| Vocational certificate | 8 (27%) | 0 |
| Bachelor's degree or higher | 9 (30%) | 4 (40%) |
| Occupations | | |
| Student | 6 (20%) | 0 |
| Full-time private employee | 12 (40%) | 0 |
| Government officer | 0 | 2 (20%) |
| Freelance | 8 (27%) | 6 (60%) |
| Agriculture | 1 (3.3%) | 0 |
| Housewife | 0 | 2 (20%) |
| Having depression at the most recent assessment | | |
| PHQ-9 score ≥ 10 | 5 (17%) | NA |

Data in number(s) (%) or median (interquartile range); NA not applicable

*Caregivers included eight biological mothers, one biological father, and one stepmother.



Internalized stigma also emerged from IDIs. A male participant grew up in a small village and had never disclosed his HIV status to anyone. He avoided joining school activities that required overnight stays. He feared taking medication outside his house when others were around.

*"I usually denied joining school camping activities. Many classmates thought I was unhealthy, as I said I needed to take medication. Most of the time I stayed home on my own." [Group B: R-03, male]*

Another participant shared feelings about himself, while he sometimes felt like he was easily tired and lacked energy.

*"Sometimes I get upset with myself and have self-pity. However, usually I just let it go. There is no direct effect on my performance at work. It is like I cannot work as much as I would like. While I see others work harder, I feel tired and desperate." [Group A: R-02, male]*

A similar experience was mentioned by another female participant. When she felt overwhelmed with her daily responsibilities, she thought of HIV as a barrier to being healthy and strong.

*"I feel a bit tired, as I need to think about taking medication every day. Yes, I am exhausted. I may not be as tough or as strong as I should be. I feel a lack of energy sometimes, and… it might be due to my studies, my work, and also… my HIV. I have always been so skinny. My friends said they were afraid that they could break my bones when we played together. My weight has never reached 50 kilos. I think it can be due to HIV." [Group B: C-02, Female]*

## Anticipated stigma

YPHIV stated their expectations that something might happen if others knew they had HIV. The fear had been with them since childhood after they became aware of their HIV status. A young male claimed that he had never disclosed his serostatus to anyone due to the fear of being discriminated against.

*"I have never told anyone about my HIV. I was afraid that they would go away and end their relationship with me. If they heard the word "HIV," they might think I am disgusting." [Group B; C-01, female]*

Another participant shared her childhood experience. She was afraid of having no one to play with. She kept this to herself and had never talked to anyone about this fear.

*"When I was young, I didn't feel OK. I was afraid that others would know I had HIV. Like my friends, they might not play with me anymore if they knew. I have never shared my fears with anyone, whether at home or in the clinic. I never want to talk about it. [Group B: C-07, female]*

Not only did YAPHIV have such fear(s) about HIV disclosure, but caregivers also anticipated being stigmatized and discriminated against in healthcare settings that they were not familiar with. A caregiver mentioned his concern about privacy if they needed to attend HIV services in the hospital in the area where their family resided.

*"If we need to go to the hospital near where we live, I am concerned about the safety of information. I mean our privacy, my son's and mine. We might see someone we know, and they might come up with questions. We don't want them to know about his medication." [Group C: C-05, male]*

### Enacted stigma and discriminated experiences

YPHIV described enacted stigma in their school life, mostly due to prejudice and stereotypes related to their physical appearance (e.g., visible scars from past infections, short stature). A female participant shared her memory in primary school.

> *"I have never taken off my long-sleeved clothes at school. Some of my classmates would call me "Gecko" because of the pigmented scars on my limbs. I don't get mad at them; I just don't want to be teased." [Group A: R-04, female]*

A caregiver shared her suffering when she took her daughter to a hospital after a road accident. When knowing about the antiretroviral medication she had to take, she observed that all providers changed their service behaviors.

> *"Once my daughter had an accident, and I took her to the emergency room. She needed to be hospitalized and could not eat or drink anything. The doctor asked if she needed to take any medication for any medical conditions. I said she was on ART, and she needed to take her pills daily. After that, no providers came near my daughter. There was no wound dressing, no blood pressure measurement, and no one even walked in to monitor her symptoms. I was devastated. Why did this happen to us? It should not be like that, you see." [Group C: R-01, female]*

Even before they disclosed their HIV status, some young people living with HIV (YPHIV) reported being treated differently from other children, which sparked their curiosity since they were unaware of what HIV was. A female participant shared her childhood experiences in a big family when she was stopped from sharing utensils or joining fun activities with others. She did not understand, but as a child, she needed to follow adults' orders.

> *"I lived with my grandma and many relatives in the big family. We got together, my cousins and me. I was not allowed to do certain things with other kids my age, like sharing a glass of water. The adults said I was sick. I thought I was healthy. I kept wondering, but I could not speak up for myself." [Group A: R-01, female]*

Some participants were old enough to understand what HIV was when the diagnosis was made. A female participant stated that her grandmother always emphasized that she could not have sex because of her HIV. She memorized her grandma's words and felt bitter about her life.

> *"My grandma said I cannot have a boyfriend; I cannot have sex with anyone. I should not have a partner, as my disease is contagious. I was like… oh! I should not start a relationship. She kept telling me since I was diagnosed with the disease." [Group B: C-01, female]*

### Transition experiences

We found various transition experiences during IDIs. Participants in group A had direct experience of getting a fresh start in new clinics. Most of them stated that they lacked the confidence to present themselves to an adult clinic with many clients, but they knew they had to. Some were satisfied, while some were not. Some YPHIV returned to pediatric clinics, while some decided to quit and did not show up anywhere. We grouped their report into 3 themes.

### Hesitation to navigate care in adult clinics and feeling unprepared

As most transitions occurred almost immediately following the change in the healthcare coverage scheme of each patient, without recognition or notice for many YPHIV. When one reached age over the cut-off or started working where social



security health coverage was provided, they would need to transition out to the adult clinic, which could be in the same or different hospitals. YPHIV were unaware of the transition until shortly before it occurred. Many were reluctant to leave and/ or unsure of what to expect in the new clinic. A participant shared her feelings when she had to attend the new clinic. She struggled at first but finally was able to manage and has continued in care since.

*"I was worried and wondered whether I would be OK or not. I did not know where to start the first time I went there. I was not used to it. It was a new place I had never visited. I had to get through it. I did everything by myself, and finally, I was done. It took some time until I felt comfortable and liked settling down in that clinic." [Group B; C-02, female]*

YPHIV lacked self-advocacy skills, which limited their ability to negotiate care needs or explain health conditions and medications. A participant who grew up in an orphanage was transitioned when she relocated to attend high school in another province where HIV was not prevalent. She mentioned her feeling unsure how to talk to the doctor and tell him about her treatment history.

*"He treated me nicely at that hospital. When he asked how I was, I didn't know what to say. He kept asking when I started my meds. I said I didn't know. People told me I started when I was young, but I recalled taking meds when I was a grown-up, not a long time ago. I was doubtful and not sure about it." [Group B: R-04, female]*

**Perceived loss of support and demotivation from being in care**

YPHIV stated that they had received support from providers in the pediatric clinic from when they were young, including materials such as toys or books, financial support including reimbursement for travel and school fees, and intangible resources such as emotional support, affection, or information. Some YPHIV had negative experiences when visiting health facilities outside the HIV clinic with whom they were familiar. They mentioned untoward service behaviors, unpleasant words, and discriminatory manners of unfriendly providers. Those experiences discouraged them from leaving the pediatric clinic.

In busier adult clinics, YPHIV felt insecure and discouraged from being in care. A participant stated his fear of seeing many people in the hospital.

*"I was assigned to another hospital after I started working. I did not go then. I was afraid to see people outside of this clinic. I did not like it. I used to go there with Grandma. The nurses talked to us badly." [Group B: C-07, female]*

Another participant decided to come back to the pediatric clinic, as he was unsatisfied with the service in the adult HIV clinic. He expected to hear more questions and receive information about his health condition.

*"I don't think they know anything about me. The doctor only signed the prescription and told me to go home. I felt like providers in the pediatric clinic had more knowledge, and they knew more about me." [Group A: R-02, male]*

A caregiver who attended the adult HIV clinic in the same hospital described the difference between clinics. She was dissatisfied with the services and did not think her son would be mature enough to face a similar situation.

*"In the adult clinic, besides the change of clinic staff, he would have to meet new doctors at each visit. He would need to be responsible for himself. They change nurses and doctors all the time. I don't know. For me, sometimes they changed my medication or made a new diagnosis. I did not understand why. I don't dare to ask, as they might think I was picky." [Group C: C-02, female]*

## Anticipated stigma and social problems

As our participants got HIV through vertical transmission, they had experienced HIV-related stigma since childhood. Internalized stigma in YPHIV was not associated with any of their behaviors. Nevertheless, the facets of their lives, including health condition, complicated family dynamics, parenting received, and socioeconomic status, limited their confidence in moving on in their lives. They learned as children that they belonged to a special group of people with a label that was not socially accepted.

A YPHIV who was a college student expressed her concern about meeting friends or people whom she knew while visiting an adult clinic where patients with various diseases sit together in the waiting area.

*"I once met a friend in the hospital, and she asked why I was there. I said I was sick and needed a doctor."* [Group B: C-02, female]

Another young lady who had just started working in an office shared her problem using the medical certificate with the clinic name printed on the header.

*"When I left the office for a clinic visit, I needed to bring back a medical certificate. I had to white out the clinic name, as it was an "HIV clinic." Unlike here, I could recall the clinic named Integrated Youth Service, which did not allow others to know what kind of services were provided."* [Group B: C-04, female]

Usually, young people with active lifestyles have a low tolerance for slow services. They have various educational or employment obligations that typically need to be managed around clinic visits. According to FGD participants, FGD-1 mentioned their preference of less than 1–2 hours of waiting, while FGD-2 said they could accept half a day in the clinic. In real situations, HIV services in some hospitals take a whole day from blood drawings to receipt of medication.

In the FGD, a participant brought up an issue that was supported by others. Longer durations in the clinic meant they would have unexplained absences from their workplace, class, or community, creating suspicion.

*"We feel uncomfortable if we have to wait long, like more than one hour; it is not OK."* [FG-1, female]

Another participant described more of their experiences at the clinic after the transition. He shared it as a fact that was not associated with any suffering.

*"What makes us feel comfortable in the clinic is rapid service. I mean half a day, like at the pediatric clinic I used to attend. Now I need a day off work to take medication. Say we arrive at the hospital at 9 or 10 AM; we need to queue, get screened, and sit in the waiting area. They will weigh us and tell us to wait for the doctor. Sometimes we will be called around 1 PM after lunch. Yes, we can go to purchase lunch and come back. Then we can talk to the doctor for around 5 minutes and go to wait again at the pharmacy."* [FG-2, male]

Since the Covid-19 pandemic, many hospitals established remote medical consultations and mailing services, which were useful for YPHIV and other PLWH who were in good health, so they did not have to go to the clinic or pharmacy as often.

*"I was only concerned about the waiting duration, and now it is much better. I have applied for the hospital mailing service. My meds are sent to my home every 3 months if I pay 100 baht before leaving the clinic. I got 2 appointment cards, one for the next blood test and another for a meeting with the doctor. I have never missed it, and there have been no problems so far."* [FG-1, male]

### Facing challenges and moving on

Not everyone has had untoward transition experiences. Some YPHIV overcame challenges and settled in adult HIV care without attrition. A male participant recalled his experiences at his first visit to an adult clinic. He received an unwelcome greeting from a nurse who upset him, but later he accepted the new environment and managed his later visits well.

> *"At first, I met with a moody nurse, and she was furious at me. On that day I forgot to bring my glasses, and I could not see the sign on the front. I walked into the room, and she shouted at me, "Do not come in." She pointed to the sign on the doorway. "Don't you see that sign?" I was upset. She could have said something like "Please wait outside for a moment; we are preparing our services." I would feel better, but she blamed me for not seeing the sign. After that, I always brought my glasses, and there was no problem. Everything is OK now." [Group A: C-01, male]*

### Transition needs

When asked about their transition needs, YPHIV did not focus on the clinic services. They mentioned processes before and after transition, which might be easier to modify. We categorized the needs of YPHIV and caregivers who remained involved in clinic attendance of youth into 3 themes as follows.

### To learn about transition beforehand

Most indicated that they wanted to learn in advance, with at least 1 or 2 visits before the transition occurred, to be well-prepared. A female participant stated that she wanted to hear about the future transition before it would happen.

> *"Two visits before going is the best timing. Yes, I mean around 6 months. You can talk to us before that, but we will forget. It is beneficial to know what will happen in the future, what will happen after you get a job. At that time, I knew that I had a social security scheme, but I did not realize that it would affect ART and clinic attendance." [FG-2, female]*

A young man stated that he wanted to learn more about transition beforehand to help him mentally prepare.

> *"I wish they could tell me in advance. One month is OK so that I can prepare myself. [Group A: R-03, male]*

Besides knowing about the transition plan, YPHIV also mentioned their need for HIV-related knowledge and wanted to learn more before moving out of the pediatric clinics. This requirement was reflected by a YPHIV who has transitioned to an adult clinic.

> *"We need to learn more about HIV, especially how it could transmit to others, before we leave. Yes, we have heard it since we were young, but we might have forgotten the details. If we do not know it before moving out, we might end up knowing nothing whatsoever. No one in the new clinic will advise us or talk to us about transmission. There was no supportive staff whom we could ask or chat with." [FG-2, female]*

### To be guided and stay connected

Almost every YPHIV stated that they did not know where to start. They mentioned including guidance on accessing care at the new clinic and having a person accompany them to reduce transition-related anxiety.

*"I want someone to walk me to the new clinic. I do not know where to go. The room and the building are different."* [Group A: R-04, female]

The idea of peer guidance emerged in the FGD. A female participant proposed that she could guide and provide some advice about the service and clinic to other youth who planned to go to the same hospital.

*"If we have already transitioned out, we can advise one another who follows the same path, like telling them what will happen at the new hospital. They can be well-prepared for seeing unkind providers and unfriendly nurses. It is unlike the pediatric clinic where we are familiar. If we know what it will be like, we can accept it easily. We might be discriminated against in the adult clinic; that is what we need to bear."* [FG-2, female]

YPHIV indicated their need to stay connected with the pediatric clinic team. Since they had bonded with pediatric providers for many years, they still wanted to be a part of the network with providers who knew who they were, listened to them, and offered the most intangible support.

*"We want to be followed. Yes, it is like a nurse calling us to ask how we are doing after moving to the new clinic. How do we feel? What happened there? She could call by phone; it is such an effortless thing, right? We can share with her what is going on and ask her questions we have."* [FG-1, male]

### Extended time in pediatric care

The median age at transition of YPHIV in this study was 23 years (IQR 20–24). We learned from some participants that they had been critically ill in the past. Now with delayed growth and development, they remain under care in pediatric clinics. We found many of them were less mature than their chronological age. Whether with or without profound neurocognitive impairment, according to a meta-analysis, impaired cognitive function was reported in individuals with perinatal HIV when compared to an uninfected population of similar age [18]. It limits their ability to comprehend and be capable of managing health service-related tasks in the adult clinic where parents or caregivers were not allowed. A young man confessed that he still needed his mother to be with him when going to an adult clinic for the first time, even though he could travel anywhere and come to the pediatric clinic by himself. The presence of clinic staff was comforting to him, but he felt it might not always be possible.

*"If I am 24-25 years old, I think I can go by myself. At first, maybe with my mom. No need for clinic staff to accompany me, but I would love it if it were possible."* [Group A: C-03, male]

The rationale for increasing age at transition was discussed in the FGD and mentioned during IDIs. YPHIV felt that after graduating they needed some time to settle themselves, like getting a permanent job, before being ready to handle the health-care transition process. A young lady proposed that care for adults with perinatal HIV remain in pediatric clinics up to age 30.

*"Now you said the age cutoff is 24 years, right? It is too soon, in my mind. I think it should be around 30 to be optimal. Twenty-four is still young, I will need someone to accompany me, but if I were 30, I could go by myself."* [Group A: C-05, female]

### Discussion

We found that more than half (60%) of YPHIV in this study had a low level of internalized stigma; the median stigma score was 14, which fell in the low range (< 16). Anticipated negative thoughts and negative thoughts toward self were



common. Anticipated and enacted stigma, as well as discrimination experiences, were revealed during IDI. Even though the national HIV treatment guideline has included the transition guidance, the outcomes remained unfavorable as evidenced in previous reports [7,8,15,19]. Our study findings contributed to a better understanding of the challenges faced by YPHIV. Transition experiences included hesitation to navigate care in adult clinics and feeling unprepared, perceived loss of support and demotivation from being in care, anticipated stigma, and social problems. Most of those who were transitioned out reported facing challenges and later being able to move on. YPHIV indicated their need to learn about the transition beforehand, to be guided to the new clinic while staying connected to their original clinics, and to extend time in the pediatric clinic.

The presence of HIV-related internalized stigma was not surprising. A larger study among 93 Nigerian youth with HIV (median age of 19.5 years) reported the presence of internalized stigma in 62% [20]. They found more negative self-image (stereotypes about HIV) in the younger age group (10–19 years) and more personalized stigma in the older age group (20–26 years). In our study, we enrolled selected participants who were willing to talk and share experiences; at the median age of 23 years, both components of internalized stigma were present. However, the study's low level of internalized stigma may not accurately reflect the broader population of YPHIV in the country, given their varied social contexts.

A US study gathered data from a survey in which 95 PLWH described that internalized or self-stigma occurs when one absorbs negative messages and stereotypes about HIV, comes to personalize them, and applies them to themselves [21]. The absorption and personalization of HIV-related messages to themselves were applied to our study participants. As they acquired HIV from vertical transmission, having HIV was not related to their health behaviors. It was more like individuals who were born with an anomaly that was not accepted, leading to self-stigmatization [22]. Negative thoughts towards self could even be present prior to disclosure to their serostatus while growing up. Nevertheless, feeling ashamed of having HIV and perceiving that having HIV was due to their bad karma could present in their mind as a child growing up with HIV. These are transferred beliefs from society, such as that seen in an Indian study. Here, mothers who gave birth to a child with an anomaly believed that it was because of bad karma, consequences of their deeds in the past life [23]. Many YPHIV grew up without the presence of biological parents; experiences of neglect could be associated with perceived stigmatization. A study among orphans in Tanzania reported a correlation between neglect or maltreatment by caregivers and mental health or behavioral problems moderated by perceived stigmatization [24]. Anticipated negative thoughts could be internalized from adults' expressions since they were young, not from direct experience or their perception while living with HIV.

The critical consequence of HIV-related stigma is non-disclosure, as YPHIV anticipated being discriminated against or the relationship ending if their serostatus was revealed [25]. To avoid labeling and being uncomfortable communicating HIV status to others, many decided to keep it secret and sacrifice an opportunity to receive social support. This could lead to disclosure-related stress, feelings of insecurity, and loss of support during the healthcare transition.

We found that YPHIV hesitated to navigate care in adult clinics independently. Since they started ART in the pediatric clinic as children, the clinic staff and their caregivers would help them receive medical care by managing all patient services. Most of the time they did not learn how to do things by themselves. This is supported by findings from a study among youth in Thailand, Malaysia, and Vietnam, where half of the participants reported feeling unprepared for transition [19]. These findings were like a US study that reported that adolescent clinic staff did almost everything in the service system for youth, which hampered their ability to independently navigate adult clinics [26]. Study participants mentioned perceived loss of support during transition out. A study among YPHIV in Uganda noted that social support from caregivers and family was required by young people [27]. It helped them to gain self-confidence and motivated them to remain in care. As YPHIV were accustomed to receiving supportive care in pediatric clinics, extended support from healthcare providers in HIV clinics beyond medical treatment was expected and considered essential to guide their treatment-related decisions.

Our finding that YPHIV wanted someone to accompany and guide them to an adult clinic was not novel. This was in line with a study in the US with adolescents living with HIV requesting that clinic staff attend their first appointment at an adult clinic [26]. One thing was that young people did not know how to get started and direct their care as they expected. A study proposed a strategy for developing connections between pediatric and adult clinics to increase successful transition by having adult clinic staff meet with youth before continuing their care after the transition [26].

YPHIV mentioned their need to stay connected with the pediatric clinic after the transition. Providing continuity through the transition period is essential for YPHIV. A previous Thai study documented multiple non-clinic challenges that influenced the capacity of YPHIV to continue in care, including socioeconomic issues, family instability, and education [28]. Those challenges and difficulties of transition could be shared with and supported by the pediatric clinic team with whom they were familiar. Staying connected also referred to engaging in their existing peer network, as mentioned in another Tanzanian study that the fear of losing their peer network was a barrier to transition [29].

Extended time in pediatric care was critical to caregivers who were interviewed in this study, while YPHIV did not take it seriously. Caregivers were worried about the youth's ability to navigate themselves in adult clinics with different environments. While family support was identified as a facilitator of transition [29], most services in adult clinics usually did not allow caregivers' involvement. Chronological age was insufficient to measure maturity when deciding on a healthcare transition [27]. YPHIV were like youth with other chronic diseases, with some delay in both physical and mental development. Some studies mentioned youth-unfriendly environments and adult providers lacking skills in dealing with young people, which hindered a streamlined transition process [26,28]. While some YPHIV could move on with resilience, each YPHIV was different, and the decision-making to transition was to be made on an individual basis rather than using a rigid cut-off.

Our study was among a few qualitative studies in this region focusing on healthcare transition. We revealed the influence of HIV-related stigma on the transition journey of YPHIV, which provided a comprehensive understanding. The study limitation included selection bias, as we selected participants willing to join, which might represent the group with more positive experiences than those who did not show interest in participating. Although we conducted a multiple-method study and collected some quantitative data, the estimated sample size was to achieve qualitative data saturation and was probably too small to draw a conclusion or generalize to other young people living with HIV. The next steps in research might include gathering experiences from providers who are involved in the pre- and post-transition process to explore their perception and awareness of the current situation. Thorough understanding of the barriers and facilitators to effective transition would allow improvement of the process.

## Conclusions

Our study revealed that internalized HIV-related stigma in Thai YPHIV might affect the transition readiness and outcomes. A variety of transition challenges were related to inadequate preparedness, perceived loss of support, anticipated stigma, and social problems. The findings will be shared with HIV care policy makers, as there is a need for practical and implementable healthcare transition guidelines or training for providers on the essential components, including how to make individualized transition plans, counseling on management of internalized stigma, and follow-up strategies after transition. A better prepared YPHIV to live with HIV through adulthood and maximize retention would lead to a better HIV treatment outcome and ensure quality of life for all.

## Supporting information

**S1 File. Demographic data.**
(XLSX)

**S2 File. Codes and excerpts.**
(XLSX)



## Acknowledgments

We thank all YPHIV who took part in this study and the study staff who managed the interviewers and FGD sessions, namely, Wanvisa Taweehorm, Nongyow Wongnum, Jinnaphak Seesuksai, Pattama Deeklum, Poramitta Padungkiattisak, and Rachaneekorn Nadsasarn, and all clinic staff at both sites who helped with the recruitment, participants' scheduling, and management of the study visit.

## Author contributions

**Conceptualization:** Linda Aurpibul, Wipaporn Natalie Songtaweesin, Thanyawee Puthanakit.

**Data curation:** Linda Aurpibul, Chutima Saisaengjan, Supunnee Masurin, Thanyawee Puthanakit.

**Formal analysis:** Linda Aurpibul, Chutima Saisaengjan, Wipaporn Natalie Songtaweesin.

**Funding acquisition:** Thanyawee Puthanakit.

**Investigation:** Linda Aurpibul, Chutima Saisaengjan, Supunnee Masurin.

**Methodology:** Linda Aurpibul, Tulathip Suwanlerk.

**Project administration:** Chutima Saisaengjan, Supunnee Masurin, Tulathip Suwanlerk.

**Resources:** Chutima Saisaengjan, Wipaporn Natalie Songtaweesin, Supunnee Masurin, Tulathip Suwanlerk, Thanyawee Puthanakit.

**Supervision:** Linda Aurpibul, Wipaporn Natalie Songtaweesin, Tulathip Suwanlerk, Thanyawee Puthanakit.

**Validation:** Linda Aurpibul, Tulathip Suwanlerk.

**Writing – original draft:** Linda Aurpibul.

**Writing – review & editing:** Chutima Saisaengjan, Wipaporn Natalie Songtaweesin, Supunnee Masurin, Tulathip Suwanlerk, Thanyawee Puthanakit.

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
