## [Decision Letter · Decision Letter 0]

8 May 2025

Thank you for submitting your manuscript to PLOS ONE. After careful consideration, we feel that it has merit but does not fully meet PLOS ONE’s publication criteria as it currently stands. Therefore, we invite you to submit a revised version of the manuscript that addresses the points raised during the review process.

We look forward to receiving your revised manuscript.

Kind regards,

Ivan Alejandro Pulido Tarquino, MSc

Academic Editor

PLOS ONE

“This study was supported by grants to amfAR from ViiV Healthcare, and the U.S. National Institutes of Health’s National Institute of Allergy and Infectious Diseases, the Eunice Kennedy Shriver National Institute of Child Health and Human Development, the National Cancer Institute, the National Institute of Mental Health, the National Institute on Drug Abuse, the National Heart, Lung, and Blood Institute, the National Institute on Alcohol Abuse and Alcoholism, the National Institute of Diabetes and Digestive and Kidney Diseases, and the Fogarty International Center, as part of the International Epidemiology Databases to Evaluate AIDS (IeDEA; U01AI069907).”

Additional Editor Comments:

Dear Dr. Aurpibul,

Thank you for submitting your manuscript entitled "HIV-related stigma, healthcare transition experiences in young people with perinatal HIV in Thailand." We appreciate the valuable contribution your work makes in addressing the transition of care and HIV-related stigma among young adults with perinatally acquired HIV in Thailand, an area that is often overlooked in the literature.

The manuscript has undergone peer review, and we would like to bring to your attention some points raised by the reviewer that will need to be addressed prior to considering the manuscript for publication.

1. Clarity and Framing of Findings in Light of Sample Size

The reviewer highlighted concerns regarding the presentation of quantitative figures within a primarily qualitative study (e.g., reporting percentages such as “60% had low levels of stigma”). Given the small sample size (n=20), such statistics can be misleading and may imply a level of precision not supported by qualitative methodology. We recommend rephrasing these statements to reflect qualitative trends or patterns without numeric quantification unless these are triangulated with validated scales.

2. Interpretation and Generalizability

The reviewer has pointed out that comparisons with studies involving larger sample sizes (e.g., references 16 and 17 with >90 participants) should be made cautiously. Please consider adding a brief reflection on how the limited sample size may affect the transferability of findings and how these limitations were managed methodologically and interpretatively.

3. Limitations Section

It is very important to explicitly highlight the limitations of the study alongside its strengths. Doing so will not only help contextualize the findings, especially those that may appear limited due to sample size or scope, but also provide a more comprehensive understanding of the study's contribution. We encourage you to reflect on methodological constraints, such as sample size and participant selection, and to discuss how these may impact the transferability or generalizability of the results.

We invite you to submit a revised version of your manuscript that addresses these comments in detail. Alongside the revised manuscript, please include a point-by-point response to each reviewer comment.

We look forward to receiving your revised manuscript.

Thank you again for your important work.

Kind regards

Ivan Alejandro Pulido Tarquino

Reviewers' comments:

Reviewer's Responses to Questions

**Comments to the Author**

1. Is the manuscript technically sound, and do the data support the conclusions?

Reviewer #1: Yes

Reviewer #2: Partly

2. Has the statistical analysis been performed appropriately and rigorously?

Reviewer #1: Yes

Reviewer #2: Yes

3. Have the authors made all data underlying the findings in their manuscript fully available?

Reviewer #1: Yes

Reviewer #2: Yes

4. Is the manuscript presented in an intelligible fashion and written in standard English?

Reviewer #1: Yes

Reviewer #2: Yes

Reviewer #1: This is an intersting article and fairly well wriiten. However it maybenefit from a tighter and shorter draft. Otherwise may i commend for awork well done. May benfit from a reviewof the English. congrats

Reviewer #2: I want to appreciate the authors for putting together this paper and highlighting an important but a neglected part of HIV care services.

Title and the abstract

• The title is not very descriptive. Please see two suggestions:

o Understanding the challenges of healthcare transition in the context of HIV-related stigma for young adults with perinatal HIV in Thailand.

o A study of HIV-related stigma and its influence on the healthcare transition journeys of young individuals with perinatal HIV in Thailand

• Overall, the abstract needs clarity.

• This statement (with numbers) cannot be part of the result of a qualitative study: Twelve YPHIV (60%) had low, six (30%) had medium, and two (10%) had high levels of internalized stigma. Please rephrase.

• If there were two or three groups, there is no mention of the findings from the interaction with care givers.

• Revise the result and start each finding by clearly mentioning which group it refers to.

• The conclusion section of the abstract should contain one or one and a half sentences to summarize the findings, and then it should end with a recommendation. Please revise.

• Is developing “individualized transitional plans” practically possible, as mentioned in the concluding sentence?

Introduction

• In the last paragraph of the introduction, please write YPHIV in full.

Methods

• What is a multiple-method study?

• Was the FGD done with each of the groups?

• No description of the Thai Internalized HIV related Stigma Scale brief (Thai-IHSS brief)

• Mention inclusion and exclusion criteria; clearly mention that the participants were positive because of vertical transmission

• Mention more about how participants were approached and recruited

• What is PHQ-9 scores?

Results

• With such a small sample size, there is limited justification for having Tables 1 and 2.

• The two studies mentioned in the discussion (references 16 and 17) had a sample size of 90 plus.

• The 17% depressive symptom statement should be deleted simply because the sample size is too small and also because the study is primarily a qualitative study.

• Lines 146-151 on printed page 9 should be deleted.

• When quoting and describing the quote, clearly mention which group it is coming from.

Discussion

• The overall discussion is well written, covering critical areas of the lives of young people with perinatal HIV.

• Please write YPHV in full in the opening paragraph.

• Please add a few sentences about the limitations of the stu

**Do you want your identity to be public for this peer review?** For information about this choice, including consent withdrawal, please see our Privacy Policy

Reviewer #1: No

Reviewer #2: **Yes: ** Arshad Altaf

---

## [Author Response · Author response to Decision Letter 1]

9 Jun 2025

Response to reviewers’ comments

Response: We have rechecked the manuscript style.

“This study was supported by grants to amfAR from ViiV Healthcare, and the U.S. National Institutes of Health’s National Institute of Allergy and Infectious Diseases, the Eunice Kennedy Shriver National Institute of Child Health and Human Development, the National Cancer Institute, the National Institute of Mental Health, the National Institute on Drug Abuse, the National Heart, Lung, and Blood Institute, the National Institute on Alcohol Abuse and Alcoholism, the National Institute of Diabetes and Digestive and Kidney Diseases, and the Fogarty International Center, as part of the International Epidemiology Databases to Evaluate AIDS (IeDEA; U01AI069907).”

Response: We have included this change in the cover letter.

Response: The data contains potentially identifying and sensitive patient information. We would choose to state that the data sets available upon request. We have uploaded two supplement data files in the system.

Additional Editor Comments:

1. Clarity and Framing of Findings in Light of Sample Size

The reviewer highlighted concerns regarding the presentation of quantitative figures within a primarily qualitative study (e.g., reporting percentages such as “60% had low levels of stigma”). Given the small sample size (n=20), such statistics can be misleading and may imply a level of precision not supported by qualitative methodology. We recommend rephrasing these statements to reflect qualitative trends or patterns without numeric quantification unless these are triangulated with validated scales.

Response: We revised the results in this part. It is now read as follows.

“The Thai IHSS brief score revealed a low level of internalized stigma in the study participants (median score 14; IQR 11-17). The components with the highest scores indicating their most concern were anticipated negative thoughts (Items 1 & 2) and negative thoughts toward self (Items 5 & 6).”

We also removed the numbers in the abstracts and replaced them with the following sentence.

“The Thai IHSS brief score revealed a low level of internalized stigma in the study participants (median score 14; IQR 11-17).”

2. Interpretation and Generalizability

The reviewer has pointed out that comparisons with studies involving larger sample sizes (e.g., references 16 and 17 with >90 participants) should be made cautiously. Please consider adding a brief reflection on how the limited sample size may affect the transferability of findings and how these limitations were managed methodologically and interpretatively.

Response: We revised the 2nd and 3rd paragraphs of the discussion as follows.

“A larger study among 93 Nigerian youth with HIV (median age of 19.5 years) reported the presence of internalized stigma in 62% [16]. They found more negative self-image (stereotypes about HIV) in the younger age group (10-19 years), and more personalized stigma in the older age group (20-26 years). In our study, we enrolled selected participants who were willing to talk and share experiences; at the median age of 23 years, both components of internalized stigma were presented. However, the low level of internalized stigma in this study might not represent the larger group of YPHIV in the country.

A US study gathered data from a survey in 95 PLWH described that internalized or self-stigma occurs when one absorbs negative messages and stereotypes about HIV, comes to personalize them, and apply to themselves [17]. The absorption and personalization of HIV-related messages to themselves were applied to our study participants. As they acquired HIV from vertical transmission, having HIV was not related to their health behaviors.”

3. Limitations Section

It is very important to explicitly highlight the limitations of the study alongside its strengths. Doing so will not only help contextualize the findings, especially those that may appear limited due to sample size or scope, but also provide a more comprehensive understanding of the study's contribution. We encourage you to reflect on methodological constraints, such as sample size and participant selection, and to discuss how these may impact the transferability or generalizability of the results.

Response: We added the study strength and limitation as follows.

“Our study was among a few qualitative studies in this region focusing on healthcare transition. We revealed the influence of HIV-related stigma on the transition journey of YPHIV, which provided a comprehensive understanding. The study limitation included selection bias, as we selected participants willing to join, which might represent the group with more positive experiences than those who did not show interest in participating. Although we conducted a multiple-method study and collected some quantitative data, the estimated sample size was to achieve qualitative data saturation and was probably too few to draw a conclusion or generalize to other young people living with HIV.”

Review Comments to the Author

Reviewer #1: This is an interesting article and fairly well writ ten. However it may benefit from a tighter and shorter draft. Otherwise may I commend for a work well done. May benfit from a review of the English. congrats

Reviewer #2: I want to appreciate the authors for putting together this paper and highlighting an important but a neglected part of HIV care services.

Title and the abstract

1) The title is not very descriptive. Please see two suggestions:

o Understanding the challenges of healthcare transition in the context of HIV-related stigma for young adults with perinatal HIV in Thailand.

o A study of HIV-related stigma and its influence on the healthcare transition journeys of young individuals with perinatal HIV in Thailand

Response: Thank you for the proposed titles. We would rather go with the first one.

In the revised manuscript, the title was changed as follows.

“Understanding the challenges of healthcare transition in the context of HIV-related stigma for young adults with perinatal HIV in Thailand.”

• Overall, the abstract needs clarity.

2) This statement (with numbers) cannot be part of the result of a qualitative study: Twelve YPHIV (60%) had low, six (30%) had medium, and two (10%) had high levels of internalized stigma. Please rephrase.

Response: We have replaced the statement with the following sentence.

“The median Thai IHSS brief score was 14 (IQR 11-17) representing a low level of internalized stigma in the overall study participants.”

• If there were two or three groups, there is no mention of the findings from the interaction with care givers.

• Revise the result and start each finding by clearly mentioning which group it refers to.

Response: The qualitative findings were from YPHIV in group A, B, and caregivers in group C. We revised the results of abstracts to make it clearer as follows.

“HIV-related stigma experiences of YPHIV and caregivers were grouped into 3 themes: internalized, anticipated, and enacted stigma/discrimination. Transition experiences of YPHIV in both groups included hesitation to navigate care in adult clinics and feeling unprepared, perceived loss of support, and demotivation from being in care. Anticipated stigma and social problems were expressed by YPHIV and caregivers.”

• The conclusion section of the abstract should contain one or one and a half sentences to summarize the findings, and then it should end with a recommendation. Please revise.

Response: We revised the conclusion of the abstract as follows.

“In conclusion, we found many stigma issues started since childhood, plus collective experience while growing up. The internalized HIV-related stigma influenced the healthcare transition journey of YPHIV. Healthcare providers need additional guidance on how to manage transition in YPHIV, including individualized transition plans for those at increased risk of adverse outcomes, interventions to manage internalized stigma, and follow-up strategies after transition.”

• Is developing “individualized transitional plans” practically possible, as mentioned in the concluding sentence?

Response: Yes, it is possible for selected cases, but not for every case. We revised it to be more realistic as follows.

“Individualized transition plans for those at increased risk of adverse outcomes,…”

Introduction

• In the last paragraph of the introduction, please write YPHIV in full.

Response: We added full term. It is now read as follows.

“young people with perinatal HIV (YPHIV)”

Methods

• What is a multiple-method study?

Response: Multiple-methods study or multimethod study is a study using two or more methods in data collection in parallel but does not integrate them in any stages until analysis. We refer to the definition in the reference below.

https://www.researchgate.net/publication/235413072_Toward_a_Definition_of_Mixed_Methods_Research_Journal_of_Mixed_Methods_Research_1_112-133

• Was the FGD done with each of the groups?

Response: No, they were mixed. We revised the last sentence in the Study settings and Population to clarify this as follows.

“We invited participants who underwent IDIs in either Group A or B who were interested in participating on another separate day by appointment.”

• No description of the Thai Internalized HIV related Stigma Scale brief (Thai-IHSS brief)

Response: We added descriptions of IHSS in the Methods section as follows.

“The 8-item Thai Internalized HIV-related Stigma Scale brief (Thai-IHSS brief) was used to assess internalized stigma. It is a tool developed from survey data among Thai PLWH and had a congruence validity and discriminant power in measuring internalized stigma [14].”

• Mention inclusion and exclusion criteria; clearly mention that the participants were positive because of vertical transmission

Response: We added a phrase to indicate vertical transmission as suggested. It is now read as follows.

“We purposely recruited YPHIV aged between 18-30 years with a known history of vertical transmission…”

• Mention more about how participants were approached and recruited

Response: We have added details about approaching and recruitment in the Methods section as follows.

“Potential participants were approached by study staff in the clinics during their regular visit for HIV care or reached out by phone for those who had transitioned out. Those who were interested underwent informed consent process before study enrollment.”

• What is PHQ-9 scores?

Response: We have added the full term of PHQ-9 which is a depression screening tool routinely used in both study clinics, and the scores were obtained from medical records. It is read as follows.

“The Patient Health Questionnaire (PHQ-9) scores for depression obtained from medical records.”

Results

• With such a small sample size, there is limited justification for having Tables 1 and 2.

Response: We revised the manuscript and decided to keep Table 1 to provide the characteristics of study participants, while removing Table 2 due to small sample size as suggested.

• The two studies mentioned in the discussion (references 16 and 17) had a sample size of 90 plus.

Response: We understand that our study, which was mainly qualitative, was far too small for comparison. However, we mentioned the similarity of findings (they went in the same direction) and referred to what was previously described in their studies that was applicable to ours.

• The 17% depressive symptom statement should be deleted simply because the sample size is too small and also because the study is primarily a qualitative study.

Response: We deleted the percentage of depression from the text Results. However, depression was considered among characteristics of study participants. Thus, we presented it in Table 1.

• Lines 146-151 on printed page 9 should be deleted.

Response: Level of HIV-related stigma is a variable of interest in this study. We know that the sample size was small, but the findings is important to fulfill the whole picture of the study. We revised to make it more qualitative as follows.

“The median Thai IHSS brief score was 14 (IQR 11-17) representing a low level of internalized stigma in the overall study participants. The components with the highest scores indicating their most concern were anticipated negative thoughts (“Others may end their relationship with me if they learn that I am living with HIV”, and “Others will think it serves me right if they know I am living with HIV”) and negative thoughts toward self (“I am ashamed that I am living with HIV” and “I think that I have HIV because of my bad karma”).”

• When quoting and describing the quote, clearly mention which group it is coming from.

Response: We revised the manuscript and specified the group on each quote. Please see an example as follows.

“[Group A: C-03, male]”

Discussion

• The overall discussion is well written, covering critical areas of the lives of young people with perinatal HIV.

• Please write YPHV in full in the opening paragraph.

Response: We added the full term of YPHIV in the first sentence of the discussion as follows.

“We found that more than half of young people living with HIV (YPHIV)…”

• Please add a few sentences about the limitations of the study

Response: We added the study strength and limitation as follows.

“Our study was among a few qualitative studies in this region focusing on healthcare transition. We revealed the influence of HIV-related stigma on the transition journey of YPHIV, which provided a comprehens

---

## [Decision Letter · Decision Letter 1]

27 Aug 2025

Dear Dr. Linda Aurpibul,

Thank you for submitting your manuscript to PLOS ONE. After careful consideration, we feel that it has merit but does not fully meet PLOS ONE’s publication criteria as it currently stands. Therefore, we invite you to submit a revised version of the manuscript that addresses the points raised during the review process.

Please submit your revised manuscript by  Oct 11 2025 11:59PM. If you will need more time than this to complete your revisions, please reply to this message or contact the journal office at plosone@plos.org . A rebuttal letter that responds to each point raised by the academic editor and reviewer(s). You should upload this letter as a separate file labeled 'Response to Reviewers'.A marked-up copy of your manuscript that highlights changes made to the original version. You should upload this as a separate file labeled 'Revised Manuscript with Track Changes'.An unmarked version of your revised paper without tracked changes. You should upload this as a separate file labeled 'Manuscript'.

We look forward to receiving your revised manuscript.

Kind regards,

Ivan Alejandro Pulido Tarquino, MSc

Academic Editor

PLOS ONE

Journal Requirements:

Additional Editor Comments:

Dear Dr. Linda Aurpibul,

Thank you for your submission and for addressing the earlier reviewer comments. The manuscript is overall technically sound and well-presented.

Before moving forward, however, we kindly request that you consider the following points to strengthen the paper, as already mentioned by the reviewer:

1. Please provide additional background statistics on the two health facilities where the study was conducted, particularly in relation to the number of YPLHIV in care and key HIV indicators.

2. Clarify whether PHQ-9 monitoring is part of routine HIV care at these sites, and provide more detail on the relationship of caregivers to the YPLHIV interviewed.

3. Consider restructuring tables to separately describe (1) characteristics of the YPLHIV interviewed compared to the broader facility population, and (2) characteristics of caregivers.

4. Strengthen the opening of the discussion by defining low-level internalized stigma earlier, and expand on whether Thailand has existing transition guidelines for YPLHIV or if these need to be developed—this will help better tie into the conclusions.

5. Make the conclusion more robust by explicitly outlining the potential next steps, especially regarding training of healthcare workers involved in the transition of YPLHIV. Also include limitations as highlighted by the reviewer.

6. Ensure consistency of style (e.g., use of reported speech) and correct minor omissions.

We believe that addressing these points will greatly enhance the clarity, impact, and applicability of your study.

Please prepare and submit a revised version of your manuscript at your earliest convenience. Do not hesitate to reach out if you have questions about these requested revisions.

With best regards,

Ivan Alejandro Pulido Tarquino

Academic Editor

Reviewers' comments:

Reviewer's Responses to Questions

**Comments to the Author**

Reviewer #2: All comments have been addressed

Reviewer #3: (No Response)

2. Is the manuscript technically sound, and do the data support the conclusions?

Reviewer #2: Yes

Reviewer #3: Partly

3. Has the statistical analysis been performed appropriately and rigorously?

Reviewer #2: N/A

Reviewer #3: Yes

4. Have the authors made all data underlying the findings in their manuscript fully available?

Reviewer #2: Yes

Reviewer #3: Yes

5. Is the manuscript presented in an intelligible fashion and written in standard English?

Reviewer #2: Yes

Reviewer #3: Yes

Reviewer #2: Thank you for addressing the comments. I am cognizant of the fact that sometimes these comments lead to frustration. However, the overall objective of any peer-review is to improve the manuscript for all categories of readers.

Reviewer #3: Comments to author: This is my 1st review, however I do note that previous reviewers provided comments and many have been addressed- This is a very interesting and pertinent topic affecting families and it is an easy read.

I would recommend revising the 1st sentence in the introduction (line 53-54) to end...children born with perinatal HIV can grow to adulthood

Technically sound: Generally technically sound research methods were used. I do recommend giving some background statistics on the 2 health facilities where the study took place. Namely 1) How many YPLHIV overall were receiving HIV care at each site during the period of the study ? 2) What are the key HIV indicators at these sites in general and among YPHIV? What is the background PHQ-9 score? ( please clarify if PHQ-9 monitoring is done as part of routine HIV care?)

Also would have been worthwhile to describe the most common relationship (parent? or other) of the care giver to the YPHIV who were interviewed and FGD.

One consideration is to have 2 tables: 1 that describe characteristics of the YPHID interviewed vs the general population of YPHIV at the 2 HF. A 2nd table would focus on the care givers' characteristics.

Line 427 should be an opening statement of the discussion as it defines what Low level internalized stigma is and it should come before line 419.

Line 431- I re-emphasize that we need to know how many YPHIV were on ARV treatment at these 2 health facilities in general.

line 446- what proportion of YPHIV had care givers who were their parents...describe relationships of care givers to the YPHIV interviewed (this is related to the comment about a possible 2nd table focused on the care givers' characteristics.

The Conclusion could be made more robust: not explicitly mentioned about actual plan for training of health care workers (especially those where YPHIV are being transferred to). FYI- It is unclear whether Thailand already has transition guidelines in place or these need to be developed. This should be mentioned in the discussion in order to tie up with the conclusion.

Statistical analysis: analytic methods took into consideration that this is a qualitative study and sample size is quite small to draw generalizable conclusions

Data availability: Yes, this seems to be addressed from previous comments and author response.

Language and presentation: For consistency in writing style, recommend to use reported speech (line 365) ;

Line 510- Please correct to ....counseling on management of internalized stigma (the word "of" is missing it seems)

**Do you want your identity to be public for this peer review?** For information about this choice, including consent withdrawal, please see our Privacy Policy

Reviewer #2: **Yes: ** Arshad Altaf

Reviewer #3: **Yes: ** Charity Ndalama Alfredo

---

## [Author Response · Author response to Decision Letter 2]

22 Sep 2025

Response to reviewers’ comment

1. Please provide additional background statistics on the two health facilities where the study was conducted, particularly in relation to the number of YPLHIV in care and key HIV indicators.

Response: We have added some information in the methods section to clarify background of the study population as follows.

“We have similar cohorts of YPHIV who were initiated on HIV treatment and followed during 2005-2020 as a part of the TREAT Asia Pediatric HIV Observational Database [14]. In 2021, there were 124 and 145 YPHIV aged 18 years or older actively followed at the sites in Chiang Mai and Bangkok, respectively; their treatment outcomes during adolescent years were included in the previous publication [15].”

2. Clarify whether PHQ-9 monitoring is part of routine HIV care at these sites and provide more detail on the relationship of caregivers to the YPLHIV interviewed.

Response: We added in the method that PHQ-9 is assessed as a part of regular clinic visits in both clinics.

“The most recent CD4 and viral load, self-reported adherence, and the Patient Health Questionnaire (PHQ-9) scores for depression, which were assessed on every clinic visit, were obtained from medical records.”

Details on relationships or caregivers were added in the footnote of Table 1 as follows.

“Caregivers included eight biological mothers, one, biological father, and one stepmother.”

3. Consider restructuring tables to separately describe (1) characteristics of the YPLHIV interviewed compared to the broader facility population, and (2) characteristics of caregivers.

Response: We would like to clarify that we did not have data on facility population that could be included in this report. As the study was designed to gather experiences from YPHIV during transitional period, data collection was focused on the qualitative data. Meanwhile, the characteristic table was only presented for readers to know the demographic background of informants.

4. Strengthen the opening of the discussion by defining low-level internalized stigma earlier and expand on whether Thailand has existing transition guidelines for YPLHIV or if these need to be developed—this will help better tie into the conclusions.

Response: We revised the first paragraph of the discussion as follows.

“We found that more than half (60%) of YPHIV in this study had a low level of internalized stigma; the median stigma score was 14, which fell in the low range (< 16). Anticipated negative thoughts and negative thoughts toward self were common. Anticipated and enacted stigma, as well as discrimination experiences were revealed during IDI. Even though the national HIV treatment guideline has included the transition guidance, the outcomes remained unfavorable as evidenced in previous reports [7, 8, 15, 18]. Our study findings served as a complement to create understanding of the challenges faced by YPHIV.”

5. Make the conclusion more robust by explicitly outlining the potential next steps, especially regarding training of healthcare workers involved in the transition of YPLHIV. Also include limitations as highlighted by the reviewer.

Response: We have revised the last paragraph of the discussion as follows.

“The next steps in research might include gathering experiences from providers who are involved in the pre- and post-transition process to explore their perception and awareness of the current situation. Thorough understanding of the barriers and facilitators to effective transition would allow improvement of the process.”

Also, we added the following statement in the conclusion

“There is a need for practical and implementable healthcare transition guidelines or training for providers on the essential components including how to make individualized transition plans, counseling on management of internalized stigma, and follow-up strategies after transition.”

6. Ensure consistency of style (e.g., use of reported speech) and correct minor omissions.

Response: Thank you. We have reviewed and reconciled the reported speech presentation as suggested.

Reviewers' comments:

7. I would recommend revising the 1st sentence in the introduction (line 53-54) to end...children born with perinatal HIV can grow to adulthood

Response: We revised the end of the sentence as suggested.

“…can grow to adulthood.”

8. Line 427 should be an opening statement of the discussion as it defines what Low level internalized stigma is and it should come before line 419.

Response: We revised the first two paragraphs of the discussion as suggested. The opening statement is now read as follows.

“We found that more than half (60%) of YPHIV in this study had a low level of internalized stigma; the median stigma score was 14, which fell in the low range (< 16).”

And we also revised the second paragraph of the discussion as follows.

“The presence of HIV-related internalized stigma was not surprising. A larger study among 93 Nigerian youth with HIV (median age of 19.5 years) reported the presence of internalized stigma in 62% [18]. They found more negative self-image (stereotypes about HIV) in the younger age group (10-19 years) and more personalized stigma in the older age group (20-26 years). In our study, we enrolled selected participants who were willing to talk and share experiences; at the median age of 23 years, both components of internalized stigma were present. However, the study's low level of internalized stigma may not accurately reflect the broader population of YPHIV in the country, given their varied social contexts.”

9. Line 431- I re-emphasize that we need to know how many YPHIV were on ARV treatment at these 2 health facilities in general.

Response: We added the details on study population in the Methods section as follows.

“We have similar cohorts of YPHIV who were initiated on HIV treatment and followed during 2005-2020 as a part of the TREAT Asia Pediatric HIV Observational Database [14]. In 2021, there were 124 and 145 YPHIV aged 18 years or older actively followed at the sites in Chiang Mai and Bangkok, respectively; their treatment outcomes during adolescent years were included in the previous publication [15].”

10. line 446- what proportion of YPHIV had care givers who were their parents...describe relationships of care givers to the YPHIV interviewed (this is related to the comment about a possible 2nd table focused on the care givers' characteristics.

Response: We added the relationship between interview caregivers and YLPHIV in the footnote of Table 1. We would like to clarify that most YLPHIV were orphaned and grew up with their relatives. However, most of caregivers who were still involved in HIV care and accompanied YLPHIV to clinics, which were the target population for this study, were biological parents or comparable (like a caring stepmother).

11. The Conclusion could be made more robust: not explicitly mentioned about actual plan for training of health care workers (especially those where YPHIV are being transferred to). FYI- It is unclear whether Thailand already has transition guidelines in place or these need to be developed. This should be mentioned in the discussion in order to tie up with the conclusion.

Response: We have revised the conclusion paragraph as follows.

“The findings will be shared with HIV care policy makers, as there is a need for practical and implementable healthcare transition guidelines or training for providers on the essential components, including how to make individualized transition plans, counseling on management of internalized stigma, and follow-up strategies after transition.”

We also mentioned the presence of Thailand’s guideline and the position of our study in the first paragraph of the discussion as follows.

“Even though the national HIV treatment guideline has included the transition guidance, the outcomes remained unfavorable as evidenced in previous reports [7, 8, 15, 18]. Our study findings served as a complement to creating understanding of the challenges faced by YPHIV.”

12. Statistical analysis: analytic methods took into consideration that this is a qualitative study and sample size is quite small to draw generalizable conclusions

Response: We have addressed the limitations in the last paragraph of the discussion as follows.

“Although we conducted a multiple-method study and collected some quantitative data, the estimated sample size was to achieve qualitative data saturation and was probably too few to draw a conclusion or generalize to other young people living with HIV.”

13. Language and presentation: For consistency in writing style, recommend using reported speech (line 365);

Response: We revised the description in the part as suggested. It is now read as follows.

“Besides knowing about the transition plan, YPHIV also mentioned their need for HIV-related knowledge and wanted to learn more before moving out of the pediatric clinics. This requirement was reflected by a YPHIV who has transitioned to an adult clinic.”

14. Line 510- Please correct .... counseling on management of internalized stigma (the word "of" is missing it seems)

Response: We corrected it. Thank you.

“…management of internalized stigma,”

---

## [Decision Letter · Decision Letter 2]

14 Oct 2025

Dear Dr. Aurpibul,

Thank you for submitting your manuscript to PLOS ONE. After careful consideration, we feel that it has merit but does not fully meet PLOS ONE’s publication criteria as it currently stands. Therefore, we invite you to submit a revised version of the manuscript that addresses the points raised during the review process.

We look forward to receiving your revised manuscript.

Kind regards,

Ivan Alejandro Pulido Tarquino, MSc

Academic Editor

PLOS ONE

Journal Requirements:

Additional Editor Comments:

Dear Dr. Aurpibul,

We would like to thank you for submitting your manuscript entitled "Understanding the challenges of healthcare transition in the context of HIV-related stigma for young adults with perinatal HIV in Thailand" to our journal. The paper has now been reviewed.

Based on the feedback received, we are pleased to invite you to revise your manuscript as a minor revision. Please address the reviewer’s suggestions carefully and provide a brief, point-by-point response indicating how each comment has been considered in your revised version.

When resubmitting, kindly highlight all changes in the revised manuscript to facilitate the review process.

We look forward to receiving your revised version through the journal’s submission system. If you anticipate any delay, please let us know in advance.

Thank you again for your contribution and for the quality of your work.

Best regards

Ivan Alejandro Pulido Tarquino

Reviewers' comments:

Reviewer's Responses to Questions

**Comments to the Author**

Reviewer #3: All comments have been addressed

2. Is the manuscript technically sound, and do the data support the conclusions?

Reviewer #3: Yes

3. Has the statistical analysis been performed appropriately and rigorously?

Reviewer #3: N/A

4. Have the authors made all data underlying the findings in their manuscript fully available?

Reviewer #3: No

5. Is the manuscript presented in an intelligible fashion and written in standard English?

Reviewer #3: Yes

Reviewer #3: Please address these minor revisions:

Line 197: "..YA-PHIV.." should this be YPHIV?

Line 407-409: .."Whether with or without 407 neurocognitive impairment, their maturity may be lower than their chronological age. It limits their ability to comprehend and be capable of managing 408 health service-related tasks in the adult...". Is this a stated fact or speculation? If this is factual, please include your reference.

Line 410: "..mom.." would be more formal to use the word "mother" for this audience/publication

Line 446: "..an US.." should this be "a US study"?

Recommend accepting the manuscript when the minor revisions have been made.

**Do you want your identity to be public for this peer review?** For information about this choice, including consent withdrawal, please see our Privacy Policy

Reviewer #3: **Yes: ** Charity Ndalama Alfredo

---

## [Author Response · Author response to Decision Letter 3]

16 Oct 2025

Response to reviewers’ comments

• Regarding the data availability, we thank the reviewer for raising this important point. Due to ethical restrictions and participant confidentiality requirements approved by the Human Ethics Committee, the individual-level data cannot be made publicly available. However, de-identified aggregated data relevant to the study findings are presented in the supplemental files. More data is available upon reasonable request from the corresponding author at any time.

Reviewer #3: Please address these minor revisions:

1. Line 197: "..YA-PHIV.." should this be YPHIV?

Response: We corrected it. It is read as follows.

“Not only did YPHIV have such fear(s) about HIV disclosure,…”

2. Line 407-409: .."Whether with or without 407 neurocognitive impairment, their maturity may be lower than their chronological age. It limits their ability to comprehend and be capable of managing 408 health service-related tasks in the adult...". Is this a stated fact or speculation? If this is factual, please include your reference.

Response: This is both fact and speculation. We revised the sentence and added a reference as follows.

“We found many of them were less mature than their chronological age. Whether with or without profound neurocognitive impairment, according to a meta-analysis, impaired cognitive function was reported in individuals with perinatal HIV when compared to the uninfected population of similar age [18].”

Line 410: "..mom.." would be more formal to use the word "mother" for this audience/publication

Response: Thank you, we changed from mom to mother as suggested.

“A young man confessed that he still needed his mother…”

Line 446: "...an US.." Should this be "a US study"?

Response: We corrected it.

---

## [Editor Report · Decision Letter 3]

28 Oct 2025

Understanding the challenges of healthcare transition in the context of HIV-related stigma for young adults with perinatal HIV in Thailand

PONE-D-25-18482R3

Dear Dr. Aurpibul,

We’re pleased to inform you that your manuscript has been judged scientifically suitable for publication and will be formally accepted for publication once it meets all outstanding technical requirements.

Kind regards,

Ivan Alejandro Pulido Tarquino, MSc

Academic Editor

PLOS ONE

Additional Editor Comments (optional):

Dear Dr. Aurpibul,

Thank you for submitting your responses.

I’m pleased to see that the quality of the manuscript has improved and that your work is now ready for publication.

The editorial services team will contact you to proceed with the next steps.

Congratulations on your achievement, and I look forward to reading more of your work on this important health topic.

Kind regards

Ivan Alejandro Pulido Tarquino

---

## [Editor Report · Acceptance letter]

PONE-D-25-18482R3

PLOS ONE

Dear Dr. Aurpibul,

I'm pleased to inform you that your manuscript has been deemed suitable for publication in PLOS ONE. Congratulations! Your manuscript is now being handed over to our production team.

Kind regards,

on behalf of

Dr. Ivan Alejandro Pulido Tarquino

Academic Editor

PLOS ONE